# Inflammasomes: Mechanisms of Action and Involvement in Human Diseases

**DOI:** 10.3390/cells12131766

**Published:** 2023-07-03

**Authors:** Dimitri Bulté, Chiara Rigamonti, Alessandro Romano, Alessandra Mortellaro

**Affiliations:** 1San Raffaele Telethon Institute for Gene Therapy (SR-Tiget), IRCCS San Raffaele Scientific Institute, Via Olgettina 60, 20132 Milan, Italy; bulte.dimitri@hsr.it (D.B.); rigamonti.chiara@hsr.it (C.R.); romano.alessandro1@hsr.it (A.R.); 2Vita-Salute San Raffaele University, 20132 Milan, Italy

**Keywords:** inflammasome, interleukin 1, pyroptosis, autoinflammatory, autoimmune, neuroinflammatory and neurogenerative disorders

## Abstract

Inflammasome complexes and their integral receptor proteins have essential roles in regulating the innate immune response and inflammation at the post-translational level. Yet despite their protective role, aberrant activation of inflammasome proteins and gain of function mutations in inflammasome component genes seem to contribute to the development and progression of human autoimmune and autoinflammatory diseases. In the past decade, our understanding of inflammasome biology and activation mechanisms has greatly progressed. We therefore provide an up-to-date overview of the various inflammasomes and their known mechanisms of action. In addition, we highlight the involvement of various inflammasomes and their pathogenic mechanisms in common autoinflammatory, autoimmune and neurodegenerative diseases, including atherosclerosis, rheumatoid arthritis, systemic lupus erythematosus, inflammatory bowel disease, Alzheimer’s disease, Parkinson’s disease, and multiple sclerosis. We conclude by speculating on the future avenues of research needed to better understand the roles of inflammasomes in health and disease.

## 1. Introduction

Inflammation is crucial in the protective immune response to harmful stimuli, such as microbial insults or cell death, yet it must be tightly regulated, as excessive inflammation is implicated in numerous systemic and chronic inflammatory diseases. Inflammation is initiated when pattern-recognition receptors (PRRs) recognize pathogen-associated molecular patterns (PAMPs) and/or danger-associated molecular patterns (DAMPs). This process subsequently induces pro-inflammatory cytokine production via downstream signaling pathways. PRRs can be either membrane-bound and sense extracellular signals, such as Toll-like receptors (TLR), or cytosolic, such as nucleotide-binding domain and leucine-rich repeat-containing receptors (NLRs), absent in melanoma 2 (AIM2)-like receptors (ALRs) and tripartite motif (TRIM) receptors [1].

As we discuss in this review, certain NLRs, ALRs and TRIM receptors assemble into inflammasome complexes that regulate inflammation at the post-translational level. Typical inflammasome configurations generally consist of a sensor protein, the adaptor apoptosis-associated speck-like protein containing CARD (ASC) and pro-caspase-1 [2]. NLR sensor proteins usually contain a central nucleotide-binding and oligomerization domain (NACHT), a C-terminal leucine-rich repeats domain (LRR), and an N-terminal effector domain that can be either a Pyrin domain (PYD) or a caspase recruitment domain (CARD) in the case of NLRC4 (Figure 1) [3]. NLRP1, specifically, also contains a C-terminal CARD and a function-to-find domain (FIIND) [4]. ALR and Pyrin sensors contain an N-terminal PYD but lack NACHT or LRR domains. Instead, these sensors have a C-terminal HIN200 double-stranded DNA (dsDNA) binding domain in the case of ALR [5], and a zinc-finger domain (B-box), a coiled coil (CC) domain and C-terminal B30.2 domain in the case of Pyrin [6]. Interaction of the sensor protein with their specific activator results in the assembly of the inflammasome complex and the clustering of their PYDs [7]. This allows the recruitment of the ASC adaptor protein composed of a PYD and CARD and initiates the ASC polymerization into filaments. Cross-linking of these filaments results in the formation of a speck-like macromolecular protein complex known as the ASC speck. The exposed CARD domains of this complex subsequently recruit pro-caspase 1 through CARD-CARD interactions resulting in its proximity-induced activation. Inflammasome assembly and later activation facilitate pro-caspase-1 autoproteolysis into active caspase-1, which in turn activates the pro-inflammatory cytokines interleukin (IL)-1β and IL-18 (Figure 2) [8]. Caspase-1 also cleaves gasdermin D (GSDMD), after which the N-terminal GSDMD domain oligomerizes to form transmembrane pores that facilitate the release of intracellular contents and induce inflammatory cell death or pyroptosis. Specifically, GSDMD-mediated pyroptosis enhances the release of IL-1β and IL-18 and other alarmins, including high mobility group protein B1 (HMGB1) and IL-1α, that mediate the inflammatory response.

Inflammasomes have gained a lot of attention in the past decades due to their involvement in many inflammatory and autoimmune diseases and much research has been focused on targeting these inflammasomes as potential anti-inflammatory therapy. However, there is a lack of understanding on the biology and activation mechanisms of many of the known inflammasome complexes and, although their involvement in several human diseases is being uncovered, not much is known on the exact pathogenic mechanisms. Understanding all of this is essential for the development of new therapies for inflammasome-driven pathologies.

In this review, we will give a detailed overview of the different inflammasomes and their known mechanisms of action. In addition, we will highlight the involvement of the different inflammasomes and discuss what is known about their pathogenic mechanisms in some of the most common autoinflammatory, autoimmune and neurodegenerative diseases.

## 2. Inflammasome Structures and Mechanisms of Action

Over the past two decades, several receptor proteins have been confirmed to assemble into inflammasomes. However, the exact activation and regulation mechanisms have only been well characterized for a few of these inflammasome complexes. Here, we give a brief overview of recent advances in the activation mechanisms of several inflammasomes in order to better understand their role in human diseases.

### 2.1. NLRP1

NLRP1 was the first NLR shown to form a cytosolic inflammasome complex that specifically recruits and activates a downstream caspase-1 [9]. The human NLRP1 (hNLRP1) is encoded by a single gene, differently from its murine counterpart, which instead is encoded by three paralogues (*NLRP1a*, *NLRP1b*, and *NLRP1c*), with the latter considered a pseudogene [10]. In addition, the expression and activation of mouse NLRP1 has been mostly studied in myeloid lineage cells, such as macrophages, whereas human NLRP1 is found to be primarily expressed at the epithelial barrier, including in keratinocytes and bronchial epithelial cells [11].

hNLRP1 is the only NLR known to undergo constitutive post-translational autoproteolysis, at position Ser1213 between the subdomains ZU5 and UPA in the FIIND [12], that results in the C-terminal (NLRP1^CT^) and N-terminal (NLRP1^NT^) portions remaining noncovalently linked. Although only a fraction of the total NLRP1 protein undergoes autoproteolysis [4], this event is essential for subsequent NLRP1 activation as the released NLRP1^CT^ self-oligomerizes and assembles the inflammasome [12,13]. Besides FIIND autocleavage, hNLRP1 also undergoes N-terminal cleavage between the PYD and NACHT domains. Interestingly, while the N-terminal PYD is fundamental for hNLRP1 activity, it is not present in the mouse NLRP1 homolog [4]. Due to these differences between mice and humans, the results of mouse studies could only partially contribute to understanding the role of hNLRP1.

The hNLRP1 CARD motif can recruit caspase-1 directly, but the interaction can also be stabilized by the PYD-CARD adaptor protein ASC. Indeed, ASC is necessary for caspase-1 auto-processing caused by hNLRP1, but not for pyroptosis or IL-1 secretion [14]. An auto-inhibitory role has also recently been attributed to the region between the PYD and NACHT domains called Linker1; at steady state, the interaction between Linker1 and the FIIND silences hNLRP1 activation in auto-inhibitory complexes (Figure 2) [15]. Another mechanism of autoinhibition relies on dipeptidyl peptidases (DPP) 8 and 9. Because the CARD-containing NLRP1^CT^ can activate caspase-1, CARD-containing NLRP1^CT^ is sequestered in a ternary complex made up of full-length NLRP1 and DPP8 and/or DPP9 (Figure 2) [16,17].

Current understanding of NLRP1 inflammasome activation is largely limited to the degradation of the NLRP1^NT^ by the proteasome [18,19]. Direct activators, such as the *B. anthracis* lethal toxin, can activate NLRP1 by degrading NLRP1^NT^ via the ubiquitin ligase UBR2, which liberates NLRP1^CT^ for inflammasome assembly [18,19,20]. Indirect activators include the inhibitors of DPP8/9. As previously mentioned, DPP9 forms a ternary complex with full-length NLRP1 and NLRP1^CT^ to sequester it and prevent its oligomerization [16,17,21]. The DPP8/9 inhibitor Val-boroPro weakens hNLRP1–DPP9 interaction and indirectly accelerates hNLRP1^NT^ degradation, promoting inflammasome activation [16].

The panel of stimuli sensed by NLRP1 is expanding. For example, Bauernfried et al. discovered that NLRP1 binds directly to double-stranded RNA (dsRNA) through its LRR domain [22]. In addition, Yang et al. identified the first viral protein—tegument protein ORF45—that directly binds to and activates the NLRP1 inflammasome in human epithelial or macrophage-like cell lines without the aid of the proteasome [15]. They also showed that ORF45 induces NLRP1 inflammasome activation in human epithelial or macrophage-like cell lines. Mechanistically, ORF45 binding to Linker1 displaces UPA from the Linker1–UPA complex and induces the release of the hNLRP1^CT^ for inflammasome assembly. NLRP1 has developed the ability to sense various molecular entities or perturbations. The mechanisms by which NLRP1 senses these various modalities are more intricate than those of a promiscuous receptor, which binds to various ligands through the same ligand-binding domain and molecular mode of action. Overall, it is obvious that NLRP1 has not yet divulged all its secrets and that there is still much research to be done in this new field.

### 2.2. NLRP3

While not the first inflammasome to be discovered, NLRP3 is the most well studied due to its critical role for host immune defenses against bacterial, fungal, and viral infections [23]. NLRP3 is mainly expressed by myeloid cells (monocytes, neutrophils, macrophages, dendritic cells) but can also be found at the level of the central nervous system [24], and epithelium. NLRP3 is a tripartite protein that consists of a PYD, a NACHT domain and a LRR domain and can be activated through canonical, non-canonical, and alternative pathways (Figure 3). For canonical NLRP3 activation, macrophages must first be exposed to priming stimuli, such as TLR, NLR (e.g., NOD1 and NOD2), or cytokine receptor ligands. These ligands ultimately activate the NF-κB transcription factor, which, in turn, upregulates NLRP3 and pro-IL-1β expression, which are not constitutively expressed in resting macrophages [25,26].

Following priming, NLRP3 can be activated by diverse stimuli, including ATP, ion flux (in particular, K^+^ efflux) [27,28], particulate matter [29,30,31], pathogen-associated RNA [32], and bacterial and fungal toxins and components [1,33]. In addition, mitochondrial dysfunction, the release of reactive oxygen species (ROS), and lysosomal disruption have been proposed to be signals for the assembly and activation of inflammasomes [34]. Given that NLRP3 does not interact directly with any of these agonists and that they are biochemically distinct, it is thought that they all cause a similar cellular signal.

Most NLRP3 stimuli cause macrophages and monocytes to experience K^+^ efflux. In fact, IL-1 maturation and release from macrophages and monocytes in response to ATP or nigericin, which are now known to be NLRP3 stimuli, is mediated by cytosolic K^+^ depletion [27,35,36,37]. Additionally, K^+^ efflux alone was shown to activate NLRP3 in murine macrophages, and high extracellular K^+^ blocks NLRP3 inflammasome activation but not NLRC4 or AIM2 inflammasome activation [28,38]. It has, therefore, been assumed that diminished intracellular K^+^ levels can trigger NLRP3 inflammasome activation [28]. Recent research, however, has found small chemical compounds—i.e., imiquimod and CL097—that activate NLRP3 independently of K^+^ efflux [39]. This finding suggests that either NLRP3 inflammasome activation is caused by an event downstream of K^+^ efflux, or that K^+^ efflux-independent pathways also exist for NLRP3 inflammasome activation.

During non-canonical NLRP3 activation, cytoplasmic lipopolysaccharide (LPS) directly binds the CARD of caspase-4/5/11 (which induces pyroptosis via GSDMD) and pannexin-1, a membrane channel that releases ATP (Figure 3) [40,41,42]. This extracellular ATP activates the purinergic P2X7 receptor (P2X7R) [43], an ATP-gated cation selective receptor that forms a pore in the plasma membrane that mediates K^+^ efflux. Besides directly causing pyroptosis, the non-canonical inflammasome also induces the canonical NLRP3 inflammasome to promote IL-1β and IL-18 maturation and release [44].

Unlike both the canonical and non-canonical pathways, the alternative inflammasome pathway does not require K^+^ efflux or ASC speck formation, and does not induce pyroptosis [45]. Rather, caspase-1 activation and IL-1 maturation and secretion in human monocytes is induced by LPS stimulation alone [46]. This alternative pathway requires caspase4/5, Syk activity, and Ca^2+^ flux instigated by CD14/TLR4-mediated LPS internalization. In murine dendritic cells, prolonged LPS exposure, in the absence of any other activating signals, resulted in NLRP3-mediated IL-1β processing and secretion independent of P2X7R [47].

NLRP3 inflammasome activation is likely regulated by various post-translational modifications, with ubiquitination and phosphorylation being the most thoroughly studied [48], as well as nitrosylation and sumoylation [49].

Several groups have demonstrated that the mitotic spindle kinase NEK7 is a crucial regulator of NLRP3 inflammasome activation [50,51]. This role of NEK7 is distinct from its function in the cell cycle, as its kinase activity is not required for NLRP3 activation [50]. According to the proposed activation model, NEK7 binding induces conformational changes to NLRP3 whereby exposed PYDs can recruit ASC leading to subsequent caspase-1 activation. Nevertheless, it was recently demonstrated that another kinase called IKKβ, which is activated during priming, causes NLRP3 to be recruited to phosphatidylinositol-4-phosphate (PI4P), an abundant phospholipid on the trans-Golgi network. When IKKβ recruits NLRP3 to PI4P, NEK7—previously believed to be essential for NLRP3 activation—becomes redundant.

In the past decade, intense efforts have been put into the investigation of the mechanism of NLRP3 inflammasome activation. However, much more work is needed to understand how diverse cell signaling events are integrated to activate the NLRP3 inflammasome.

### 2.3. NLRP6

NLRP6 ensures microbial homeostasis, as shown in NLRP6-deficient mice that exhibit decreased IL-18 levels and dysbiosis, altering the composition of the intestinal microbial community. Indeed, NLRP6 is highly expressed in intestinal goblet cells and in lungs, liver, and tubular epithelium of kidneys. Moreover, it seems to have a role in the regulation of homeostasis in the periodontium and gingiva [52]. NLRP6 has an N-terminal PYD, a nucleotide-binding domain, and a C-terminal LRR (Figure 1) [52]. Gram-positive, bacteria-derived lipoteichoic acid activates the NLRP6 inflammasome by binding its LRR domain and cleaves caspase-11 through the glycerophosphate repeat of lipoteichoic acid [53]. LPS also directly binds the NLRP6 monomer via its LRR, inducing conformational changes and dimerization, which, together with ASC and caspase-1, form the inflammasome complex responsible for the maturation of the pro-inflammatory cytokines IL-1β and IL-18 [54]. NLRP6 is also involved in the anti-viral response, as seen in NLRP6-deficient mice that are more susceptible to encephalomyelitis virus infections than their wildtype counterparts [55]. This function of NLRP6 is achieved in collaboration with DHX15, which together recognize dsRNA to induce type I interferon (IFN) and IFN-stimulated gene activation via mitochondrial antiviral signaling proteins to counteract viral infections.

### 2.4. NLRP7

NLRP7 belongs to the family of signal-transducing ATPases and, to date, it has been described in humans and sheep. It has been reported that human NLRP7 is expressed in B, T, and monocytic cells, as well as in the lung, spleen, thymus, testis, and ovaries [56]. The ability of NLRP7 to form an inflammasome complex remains controversial. A study in human macrophages showed that NLRP7 can form an inflammasome complex in response to bacterial infections [57]. Specifically, mycoplasma and Gram-positive bacterial infections can activate NLRP7, which, in turn, induces IL-1β secretion. To form an inflammasome, NLRP7 requires binding and hydrolysis of ATP in its NACHT domain [58]. Moreover, complex post-translational modifications regulate its activity; namely, NLRP7 is either ubiquitinated to regulate its functions, or deubiquitinated by the STAM-binding protein to prevent its trafficking to lysosomes and its degradation [59]. Some observations also suggest that NLRP7 might have anti-inflammatory activity under certain conditions. NLRP7 can inhibit IL-1β secretion mediated by the NLRP3 inflammasome without affecting NF-κB activation required for the priming [60,61]. These features have led to the hypothesis that the interaction between NLRP7, pro-caspase-1, and pro-IL-1β may inhibit pro-IL-1β maturation. Indeed, peripheral blood mononuclear cells from hydatidiform mole patients with mutations in *NLRP7* exhibit reduced IL-1β secretion upon LPS treatment compared to healthy individual cells [62]. Whether these mutations are gain- or loss-of-function remains to be elucidated.

Altogether, these observations need to be better characterized to evaluate the involvement of NLRP7 in regulating inflammation. Moreover, the observation of an anti-inflammatory role in non-immune cells represents a good starting point to elucidate the mechanism that leads to NLRP7 inflammasome activation.

### 2.5. NLRP10

NLRP10 (also known as NOD8, PAN5, or PYNOD) is the only NLR lacking the characteristic LRR domain involved in protein–protein interactions, suggesting that NLRP10 might have an inflammasome-independent function. NLRP10 is expressed in various human and mouse tissues and cell types, including epithelial cells, keratinocytes, macrophages, DCs, and T cells [63]. However, the expression patterns seem to be cell-type- and context-dependent, and its function may vary depending on the cellular environment and the signaling pathways involved. Although the physiological role of NLRP10 has been largely uncharacterized, data suggest a role in the recognition and response to bacterial pathogens (*Salmonella* and *Mycobacterium tuberculosis*) and parasites (*Leishmania major*) [64,65,66].

Early studies showed that NLRP10 negatively regulates NF-κB activation, cell death, and IL-1β release [67], and inhibits caspase-1-mediated maturation of IL-1β [64]. Conversely, others reported normal canonical activation of NLRP3 and IL-1β production in NLRP10-deficient mouse DCs [65]. These pieces of evidence point to the possibility that NLRP10 may operate variably in different cellular environments. Indeed, Próchnicki et al. and Zheng et al. identified that the phospholipase C activator 3m3-FBS is the first trigger for NLRP10-based inflammasome assembly in colonic epithelial cells and differentiated keratinocytes [68,69]. Mechanistically, 3m3-FBS causes mitochondrial destabilization that recruits NLRP10 to damaged mitochondria, where it assembles, independently of the priming step, to form a canonical inflammasome together with ASC and caspase-1. Further research is needed to fully understand the immune and non-immune functions of NLRP10.

### 2.6. NLRP12

NLRP12 was described in 2012 as a negative regulator of the NF-κB in activated B-cell signaling, with a crucial role in controlling inflammation in both hematopoietic and non-hematopoietic compartments [70]. Indeed, we now know it is expressed in bone marrow DCs, neutrophils, macrophages, and granulocytes [71]. NLRP12 negatively regulates the canonical NF-κB signaling pathways by interacting with hyperphosphorylated IRAK1, which inhibits its accumulation. On the other hand, NLRP12 dampens the non-canonical pathway by inducing the degradation of NF-κB-inducing kinase via interaction with TRAF3 [70,72,73]. One of the consequences of suppressing NFκB signaling is that macrophages do not produce the chemoattractant factor CXCL1, negatively impacting neutrophil migration and recruitment to infection sites during microbial infections [74,75,76]. Moreover, NLRP12 negatively regulates T-cell responses, as shown by the higher production of IFN-γ, IL-17, and Th2-associated cytokines in NLRP12-deficient compared to wildtype T cells [77,78].

Besides negatively regulating immune signaling, NLRP12 has been studied as an inflammasome component. For example, during *Yersinia pestis* infection, NLRP12 activation induces the caspase-1, IL-1β, IL-18 cascade [79]. Although the NLRP12 activation mechanisms remain unknown, NLRP12 ligand generation requires the presence of virulence-associated type III secretion systems, suggesting NLRP12 activation may involve sensing damage associated with type III secretion. However, even if NLRP12 is involved in in vivo resistance against *Yersinia* infection, NLRP3 activation is required in both *Yersinia* and *Plasmodium* infections, suggesting that differential NLR activation might contribute to optimal protection and host defenses [80].

### 2.7. Pyrin

Pyrin encoded by *MEFV* is expressed largely in granulocytes, eosinophils, and monocytes. Early structural investigations of Pyrin revealed a nuclear role, indicated by the presence of a bZIP transcription factor domain and two overlapping nuclear localization signals [81]. Although full-length Pyrin is primarily found in the cytosol, later studies looking into its localization and function discovered the colocalization of the N-terminal Pyrin fraction with microtubules and the actin cytoskeleton [82].

The Pyrin C-terminal B30.2 domain is of particular significance because most familial Mediterranean fever (FMF)-associated mutations cluster there and functional data suggest that this domain is necessary for the molecular pathways causing FMF. In vitro overexpression studies demonstrated direct interaction between caspase-1 and pyrin B30.2 but others examining the impact of FMF-related mutations on the binding affinity of B30.2 to caspase-1 produced contradictory findings [6,83].

The ligand or signals that activate Pyrin have long been unknown. In 2014, Xu et al. [84] showed that Pyrin is able to sense pathogen-induced changes in the host Rho guanosine triphosphatases (Rho GTPases) (Figure 2). For example, the *Clostridium difficile* virulence factor TcdB, which glycosylates and subsequently inhibits the activity of a minor Rho GTPase called RhoA, can activate the Pyrin inflammasome [85]. When exposed to wildtype TcdB, bone-marrow-derived macrophages show a potent Pyrin-mediated inflammasome response, enhanced caspase-1 activity, and pyroptosis, which does not occur upon exposure to mutant TcdB. Inhibition of RhoA is not restricted to TcdB, as other bacterial toxins also, such as C3 (*Clostridium botulinum*), pertussis toxin (*Bordetella pertussis*), VopS (*Vibrio parahaemolyticus*), IbpA (*Histophilus somni*), and TecA (*Burkholderia cenocepacia*), can distinctly modify the RhoA switch I region domain [86,87,88]. Due to the lack of direct interaction between Pyrin and RhoA, Pyrin is believed to be activated by an indirect signal downstream of RhoA, rather than through direct recognition of specific RhoA modifications. Given that Rho GTPases regulate many aspects of actin cytoskeleton dynamics, it is, therefore, hypothesized that changes in the cytoskeleton organization might trigger Pyrin. Moreover, Pyrin activation relies on the RhoA-dependent serine/threonine-protein kinases PKN1 and PKN2, that directly phosphorylate Pyrin at Ser208 and Ser242 [89]. As a result, the chaperone proteins 14-3-3ε and 14-3-3τ interact with phosphorylated Pyrin, preventing the development of an active inflammasome and maintaining Pyrin in an inactive state. Bacterial toxins that inactivate RhoA result in decreased PKN1 and PKN2 activity and decreased amounts of phosphorylated Pyrin, which frees pyrin from 14-3-3 inhibition and promotes the development of an active Pyrin inflammasome. Studies of the autoinflammatory disorder caused by mevalonate kinase (MVK) deficiency offered more proof that the Pyrin inflammasome regulation mechanism described there is accurate. The mevalonate pathway is an important metabolic pathway that generates several metabolites, including geranylgeranyl pyrophosphate. This metabolite acts as a substrate for the geranylgeranylation of proteins, a post-translational lipid modification. RhoA is geranylgeranylated, and its translocation from the cytosol to the cellular membrane, which is required for activation, is dependent on this post-translational modification. Inhibiting the MVK pathway in bone-marrow-derived macrophages causes the release of membrane-bound RhoA and Pyrin inflammasome-dependent production of IL-1β [89]. By adding geranylgeranyl pyrophosphate, or by chemically activating PKN1 and PKN2, the synthesis of IL-1 was prevented.

Recent research has identified a previously unknown regulatory and molecular connection between AIM2, Pyrin, and ZBP1, which promotes the formation of the AIM2 PANoptosome, a multiprotein complex that includes various inflammasome sensors and cell death regulators [90].

### 2.8. NLRC4

NLRC4 was first described in 2001 as an activator and recruiter of caspase-1 upon bacterial pathogen sensing. Indeed, NLRC4 combines with pro-caspase-1 via CARD–CARD interactions to induce its processing and activation [91]. Specifically, the NLRC4 CARD interacts with the ASC adaptor protein CARD, thus linking NLRC4-ASC with caspase-1 to mediate downstream signaling [92]. Indeed, the CARD domain of ASC is necessary for recruiting caspase-1 to ASC specks, ensuring correct pro-IL-1β and pro-IL-18 proteolytic cleavage and activation [93,94,95], thus triggering proteolytic processing and oligomerization of GSDMD leading to pyroptosis [96]. NLRC4 is mainly expressed in myeloid cells, astrocytes, retinal pigmented epithelial cells, and intestinal epithelial cells [97].

NLRC4 forms an inflammasome complex with NAIP proteins, comprising three N-terminal baculovirus IAP-repeat domains, a central NACHT, and a C-terminal LRR [8], and acts as an upstream sensor of bacterial ligands in the cytoplasm. As such, the NAIP-NLRC4 inflammasome recognizes cytoplasmic bacterial ligands (mainly Gram-negative bacteria) and induces an inflammatory response via caspase-1 activation and pyroptosis. The NAIP-NLRC4 inflammasome in human and murine macrophages is activated by flagellin [98,99,100] and the type III [101,102] or type IV secretion system [103,104] proteins through direct recognition via NAIP proteins. It has been shown in mice that IFN regulatory factor 8 is responsible for the transcriptional induction of *Nlrc4* and *Naip 1*, *2*, *5*, and *6* [105]. The formation of the NAIP-NLRC4 inflammasome is quite peculiar. NLRC4 activation starts with the formation of a ligand-bound NAIP complex that changes the conformation of an NLRC4 monomer, exposing the “catalytic surface” of the active monomer, allowing it to interact with the “acceptor surface” of an inactive NLRC4 monomer. This contact activates a second monomer responsible for engaging more NLRC4 monomers, thus triggering the formation of an NLRC4 coil [7,92,106,107]. Consequently, NLRC4 oligomerization induces ASC and caspase-1 recruitment. NLRC4 inflammasome activation is finely regulated by phosphorylation and ubiquitination events [108,109].

NLRC4 was initially thought to induce inflammation by activating the caspase-1, IL-1β, and IL-18 cascade and promoting GSDMD maturation. Later data showed, however, that an artificial NAIP5-NLRC4 activator can induce the release of arachidonic acid by activating the calcium-dependent phospholipase A2 [110]. Arachidonic acid, in turn, stimulates the rapid production of prostaglandins and leukotrienes. The mechanism of arachidonic acid release and its link with NLRC4 activation is still unclear; however, more than 1000 possible targets of caspase-1 have been identified that might cooperate in NLRC4 activation to activate numerous downstream signals involved in the inflammatory response [92].

### 2.9. AIM2

Unlike other inflammasome activators, dsDNA can activate an ASC-dependent, but NLRP3-independent, inflammasome, the AIM2 inflammasome [111]. AIM2 is expressed by myeloid cells, keratinocytes, and T regulatory cells [97], and it is composed of an N-terminal PYD domain and a C-terminal hematopoietic expression, IFN-inducible, and nuclear localization (HIN) domain that senses dsDNA (Figure 1). An HIN-dependent interaction between AIM2 and dsDNA is enabled by two, high-affinity dsDNA binding folds in the HIN domain. Under homeostatic conditions, PYD and HIN form an intramolecular complex that inhibits inflammasome activation; this inhibition is relieved when the HIN domain binds dsDNA. Hence, the PYD can interact with ASC, allowing it to polymerize and thereby activate AIM2 [5,112].

The interaction between dsDNA and the AIM2 complex is independent of the DNA sequence or its origin, but the DNA must be at least 80 base pairs long to be sensed by the HIN domain [113]. Indeed, host DNA (including mitochondrial DNA, damaged nuclear DNA, and exosome-secreted host DNA released in the cytosol) and intracellular viral and bacterial DNAs released upon microbial infections, can all trigger AIM2-dependent innate immunity [114,115,116]. The AIM2 inflammasome has canonical and non-canonical activation mechanisms (Figure 2) [117]. Canonical activation, which mostly occurs during viral infections [118], is rapid and does not involve type I IFN activation [119]. By this mechanism, dsDNA is directly recognized by AIM2, triggering the formation of the inflammasome. Non-canonical activation, conversely, depends on IFN activity and is principally involved in bacterial infections [120]. Unlike canonical activation, non-canonical activation is thought to involve intracellular bacteria that escape the vacuoles and release small amount of DNA, thereby activating cyclic-GMP-AMP synthase and IFI204, which are two components of the cascade that drive IFN secretion [121]. At this point, secreted type I IFN exits the cells where it binds IFN receptors, driving the downstream activation of immunity-related GTPase family member b10 and guanylate-binding proteins, which, in turn, induce bacteriolysis releasing large quantities of bacterial DNA that are eventually recognized by the AIM2 inflammasome [122,123]. Unlike other DNA sensors involved in IFN induction, AIM2 inflammasome assembly following detection of cytosolic dsDNA drives the proteolytic maturation of IL-1β and IL-18 and the maturation of GSDMD, which induces pyroptosis [5,124,125].

To prevent cytokine overexpression and cell death, AIM2 inflammasome activation must be tightly regulated; this regulation is achieved through PYD:PYD or CARD:CARD interactions [126]. The presence of three human PYD-only (POP) genes (*POP1*, *POP2*, and *POP3*) suggests that POPs may negatively regulate inflammasomes [127,128,129]. POP1 and POP2 are broad-spectrum inhibitors that interfere with inflammasome assembly by interacting with ASC PYD. Conversely, POP3 specifically inhibits AIM2 by binding to AIM2 PYD, consequently blocking the AIM2 and ASC interaction [128].

Pathogens have evolved strategies to escape AIM2 inflammasome activation. For example, the human cytomegalovirus virion protein pUL83 by interacting with AIM2 inhibits its activation [126]. Huang et al. demonstrated that THP-1-derived macrophages infected with HCMV showed increased levels of AIM2 at early stages of infections, but 24 h post-infection AIM2 decreased to basal levels. They investigated the effect of pUL83 on AIM2 in recombinant HEK293T cells expressing AIM2, ASC, pro-caspase-1, and pro-IL-1β and found that, upon induction of AIM2 activation, the expression of pUL83 led to a drastic reduction in AIM2, pro-caspase-1, and pro-IL-1β levels. These results demonstrate that pUL83 is responsible for reducing AIM2 response leading to downstream reduction of caspase-1 and IL-1β cleavage [130]. Moreover, since AIM2 inflammasome becomes active every time it senses cytosolic dsDNA, some bacterial pathogens can escape AIM2 recognition by maintaining their structural integrity [131].

## 3. Role of Inflammasomes in Human Disease

Inflammasome activation is a crucial step in the protective immune response; however, unchecked, it may lead to chronic inflammation, which, in turn, forms a major risk for the development of autoinflammatory and autoimmune diseases. Inappropriate inflammasome activation has been implicated in the pathogenesis of many human inflammatory and autoimmune diseases. In this section, we will discuss the involvement of the different inflammasomes in some of the most common autoinflammatory and autoimmune diseases, such as arthrosclerosis, systemic lupus erythematosus, rheumatoid arthritis and psoriasis and inflammatory bowel disease. In addition, we will discuss the role of inflammasomes in neurological inflammatory diseases.

### 3.1. Autoinflammatory Diseases

#### 3.1.1. Inflammasomopathies

Inappropriate inflammasome activation results in autoinflammatory diseases called inflammasomopathies that are characterized by aberrant IL-1β and IL-18 production and excessive pyroptosis [132]. Although the various reported inflammasomopathies are often mechanistically related, the pathologies can greatly differ depending on the inflammasomes involved. The best described inflammasomopathies result from autosomal dominant gain-of-function mutations in *Nlrp3*, which drive autoinflammatory diseases known as cryopyrin-associated periodic syndrome (CAPS) [133].

CAPS is induced by uncontrolled IL-1β release and has three main clinical phenotypes: familial cold autoinflammatory syndrome (FCAS), Muckle–Wells syndrome (MWS), and neonatal-onset-multisystem inflammatory disease (NOMID). FCAS is the mildest form of CAPS and is characterized by recurrent episodes of fever accompanied by arthralgias, myalgias and maculopapular rash upon exposure to cold. Given the heterogeneity of clinical symptoms, other inflammasome genes have been implicated in this syndrome. Several case reports described mutations in *Nlrp12* [134,135,136] and *NLRC4* [137] that contribute to FCAS2 and FCAS4, respectively (Table 1).

MWS is a moderate/severe form of CAPS, which manifests with episodes of recurrent fever, rash, and arthralgia, and usually leads to hearing loss [138]. Finally, NOMID, the most severe form of CAPS, affects young infants who present with arthropathy, chronic urticaria, and central nervous system effects. To date, the Infevers database (https://infevers.umai-montpellier.fr/web/index.php accessed on 4 June 2023), a large registry for hereditary autoinflammatory disease mutations, lists more than 200 sequence variants of *Nlrp3*. These variants are classified as ‘benign’, ‘likely benign’, ‘uncertain significance’, ‘pathogenic’ and ‘likely pathogenic’. More than 100 variants are described as pathogenic/likely pathogenic, with the majority located in exon 3 (Table 1) [139]. Some genotype/phenotype correlations have been demonstrated in CAPS. For example, up to 75% of CAPS patients in North America have the L353P mutation, which is associated with a mild FCAS phenotype, while the R918Q variant is associated with late-onset hearing loss [140]. Nevertheless, several case reports described patients with the same mutations but different clinical phenotypes, suggesting that other factors might influence the disease phenotype [139].

Several *NLRC4* variants have been described that cause a severe autoimmune disease called autoinflammation with infantile enterocolitis (AIFEC) (Table 1) [141,142]; this recurrent fever syndrome shares many similarities with macrophage activation syndrome and, in severe cases, with primary hemophagocytic lymphohistiocytosis [137,141,143,144]. A major difference distinguishing AIFEC from macrophage activation syndrome and primary hemophagocytic lymphohistiocytosis, is the extremely elevated IL-18 levels found in AIFEC. Interestingly, IL-18 is also the main driver of *NLRP1*-associated autoinflammation with arthritis and dyskeratosis [145]. This very rare inflammasomopathy is caused by autosomal dominant or recessive mutations in *Nlrp1*, and is characterized by diffuse skin dyskeratosis, recurrent fevers, autoinflammation, and arthritis. Patients have high serum levels of caspase-1 and IL-18 compared to healthy controls. Currently, only five *Nlrp1* variants have been associated with this rare disease, including: A59P, R726W, L813P, P1214R and L1214L (Table 1) [146].

Besides NLR inflammasomopathies, Pyrin-related autoimmune diseases have also been described. As discussed earlier, *MEFV* mutations have been linked to FMF and pyrin-associated autoinflammation with neutrophilic dermatosis [6,83]. FMF is characterized by recurrent bouts of fever that resemble acute inflammation episodes; a severe complication is secondary amyloid A amyloidosis, which mainly affects the kidneys and is a major cause of mortality.

Current treatment options for inflammasomopathies are focused on targeting the downstream effector cytokines. For example, in the case of CAPS, anti-IL-1 therapy (anakinra, canakinumab, rilonacept) is the recommended treatment. However, since inflammation in many patients is not only mediated by inflammasome-dependent, but also inflammasome-independent cytokine production, such as TNF-α and IL-6, supportive treatment with anti-inflammatory drugs is necessary.

#### 3.1.2. Atherosclerosis

NLRP3 has been best characterized for its role in atherosclerosis initiation and progression through its role in promoting vascular inflammation. Atherosclerotic plaques show strong mRNA and protein NLRP3, ASC, caspase-1, IL-1β and IL-18 expression in macrophages, foam cells and endothelial cells (Table 2) [147]. Several studies have implicated inflammasome-derived IL-1β and IL-18 in the development of atherosclerosis [148]. Data from the Canakinumab Anti-Inflammatory Thrombosis Outcome Study (CANTOS), in which the effect of treatment with the therapeutic monoclonal anti-IL-1β antibody canakinumab on the recurrence of cardiovascular events was tested in a large randomized double-blinded trial, showed that IL-1β neutralization significantly reduced the incidence of cardiovascular events in atherosclerosis patients, suggesting a key role of IL-1β in disease activity [149].

NLRP3 is associated with cholesterol crystals inside and outside macrophages and foam cells. Indeed, cholesterol is a main driver of NLRP3 activation during atherogenesis: data from an in vitro study by Duewell et al., showed that cholesterol crystals are ingested by phagocytes and activate NLRP3 through a process that involves phagolysosomal damage [150]. Cholesterol and triglycerides also activate NLRP1 in human endothelial cells in vitro [151].

Other mechanisms of NLRP3 activation have been implicated in atherosclerosis. In human macrophages, NLRP3 activation and IL-1β secretion are induced by hypoxia, and IL-1β localizes to macrophage-rich areas with high caspase-1 and hypoxia marker expression in atherosclerotic plaques [152]. In mice, significant ATP release from necrotic cells in atherosclerotic plaques has been linked with P2X7R-dependent NLRP3 activation [153]. Dying cells in the necrotic core might also drive AIM2 activation through releasing abundant dsDNA. Pertinently, AIM2 expression was significantly increased in atherosclerotic lesions of patients, suggesting a pathogenic role in atherosclerosis [154].

Despite the overwhelming evidence of the involvement of inflammasomes in the pathogenesis of arthrosclerosis, limited clinical studies have focused on blocking inflammasomes for the treatment of atherosclerosis. A clinical study with low-dose colchicine, a drug preventing microtubule assembly and thereby disrupting inflammasome activation, showed promising results in preventing coronary artery disease; however, increasing gastrointestinal intolerance against the drug was observed [155]. Additional clinical studies are ongoing as well as studies investigating the potential of small molecule inhibitors of NLRP3. In contrast, several studies have investigated the beneficial effects of IL-1 inhibition with the above-mentioned CANTOS study demonstrating the benefits of canakinumab for the treatment of atherosclerosis [149].

**Table 2 cells-12-01766-t002:** Involvement of inflammasomes in human autoimmune and autoinflammatory diseases. Overview of the different inflammasomes and inflammasome activators involved in the pathogenesis of atherosclerosis, psoriasis, inflammatory bowel disease, rheumatoid arthritis, Sjogren’s disease, systemic lupus erythematosus, Alzheimer’s disease, Parkinson’s disease, and multiple sclerosis.

Disease	Inflammasome	Cell Type/Tissue	Activator in Human Disease	References
*Atherosclerosis*	NLRP1	Endothelial cells (in vitro)	Cholesterol Triglycerides (in vitro)	[151]
	NLRP3	Macrophages, foam cells, endothelial cells	Cholesterol Triglycerides ATP (from necrotic cells)	[147,150,152,153]
	AIM2	Necrotic lesions	dsDNA	[154]
*Psoriasis*	NLRP1	PBMCs, keratinocytes, psoriatic lesions	Psoriasin (S100A7)	[156,157]
	NLRP3	Psoriatic biopsies, keratinocytes, whole blood	CD100 IL-17, IL-22, TNF-α	[156,157,158]
	AIM2	Lesional and non-lesional skin, keratinocytes	dsDNA	[159,160]
*Inflammatory bowel disease*	NLRP3	PBMCs, colonic biopsies, intestinal mucosal cells	Intestinal microbiota	[161,162,163]
*Rheumatoid arthritis*	NLRP1	PBMCs, synovial cells	P2X4 agonist	[164,165]
	NLRP3	PBMCs, monocytes	Unknown	[166]
*Sjogren’s syndrome*	NLRP3	PBMCs, salivary glands circulating monocytes	ATP, circulating free DNA	[167,168,169]
	AIM2	PBMCs, salivary glands	Circulating free DNA	[169,170]
*Systemic Lupus Erythematosus*	NLRP3	Mononuclear cells, monocytes	Neutrophil extracellular traps, anti-dsDNA antibodies, reactive oxygen species, K^+^ efflux	[171,172]
	AIM2	Renal tissue	Neutrophil extracellular traps	[173]
*Alzheimer’s disease*	NLRP1	Monocytes, neurons	Amyloid-β K^+^/Ca^2+^ imbalance	[174,175,176]
	NLRP3	Monocytes, microglia, astrocytes	Amyloid-β	[174,177,178,179]
	NLRC4	Brain samples	Unknown	[180]
*Parkinson’s disease*	NLRP3	Monocytes, microglia	α-synuclein (Lewy bodies), reactive oxygen species	[181,182]
*Multiple sclerosis*	NLRP3	Macrophages, microglia, astrocytes, CNS tissue	Unknown	[183,184]
	NLRC4	Astrocyte-rich brain tissue, regions of demyelination	Unknown	[179]

#### 3.1.3. Psoriasis

Epidermal AIM2 mRNA and protein expression are upregulated in skin diseases, including psoriasis, atopic dermatitis, venous ulcers, and contact dermatitis [159]. In psoriasis and atopic dermatitis, AIM2 expression was increased in both lesional and non-lesional skin, and, in psoriatic lesions, cytosolic DNA triggers AIM2 activation in keratinocytes (Table 2) [159,160]; however, the source of cytosolic DNA in psoriatic keratinocytes remains unclear. In this context, autophagy was identified as a potential negative regulatory mechanism for AIM2 inflammasome activation, possibly through the elimination of damaged mitochondria (and so preventing mitochondrial DNA release) and the removal of HMGB1-DNA complexes known to activate AIM2 [185]. Interestingly, reduced autophagy was observed in keratinocytes from psoriasis patients [186]. In addition, it was recently shown that neutrophils from psoriatic patients were more prone to form NETs and that these NETs were able to induce AIM2 activation in keratinocytes in vitro [187].

NLRP1 has also been implicated in psoriasis. Increased NLRP1 mRNA expression has been detected in peripheral blood mononuclear cells isolated from psoriasis patients, and NLRP1 can be activated by psoriasin (S100A7) and dsDNA in keratinocytes, resulting in the caspase-5-dependent release of IL-1β [188,189]. Interestingly, vitamin D can suppress the caspase-5-dependent release of IL-1β in keratinocytes and psoriatic lesions. Ekman et al. also identified genetic variations of the *NLRP1* gene that were correlated with increased vulnerability to psoriasis. They reported a higher transmission of the rs878329C and rs8079034C genotype in psoriasis patients and correlated the rs878329C allele with elevated circulating IL-18 levels [190].

Finally, increased mRNA and protein expression of NLRP3 was also observed in psoriatic biopsies and this was correlated to enhanced IL-1β and caspase-1 [156]. Zhang et al. showed that the NLRP3 inflammasome in keratinocytes is activated by soluble CD100, which is increased in the sera of psoriasis patients, through its interaction with the transmembrane receptor PlxnB2 on keratinocytes [157]. In addition, the inflammatory milieu can also further aggravate psoriasis through activation of NLRP3. Both IL-17 and IL-22 were shown to activate the ROS-dependent NLRP3-caspase-1 pathway in keratinocytes, resulting in increased IL-1β secretion [191]. Moreover, a recent study by Verma et al. showed that TNF-α selectively primes the expression of pro-IL-1β, pro-IL-18 and NLRP3, but not NLRP1, AIM2 or NLRC4, in whole blood samples of psoriasis patients [158]. TNF-α inhibition has indeed been shown to be highly effective for the treatment of psoriasis. Patients treated for at least 8 months with anti-TNF-α antibodies showed normal plasma IL-1β and IL-18 levels and reduced caspase-1 activity in blood monocytes [158]. In contrast, methotrexate, a folate antagonist used for the treatment of severe inflammatory conditions, including psoriasis, did not reduce caspase-1 activity in psoriasis patients, suggesting that TNF-α inhibition is superior in reducing NLRP3-dependent inflammation during psoriasis.

#### 3.1.4. Inflammatory Bowel Disease

Dysregulation of gut homeostasis may lead to overreactive inflammatory responses resulting in different types of inflammatory bowel diseases (IBD), such as Crohn’s disease (CD) and ulcerative colitis (UC). Early reports showed increased expression and secretion of IL-1β and IL-18 by intestinal mucosal cells from CD and UC patients, suggesting the involvement of inflammasomes in the pathogenesis of these diseases [192,193]. Furthermore, single-cell analyses of mucosal tissue from IBD patients revealed an increased abundance of IL-1β^+^ macrophages, monocytes, and dendritic cells [194]. Increased activation of NLRP3 in peripheral blood mononuclear cells was observed in the majority of patients with CD, while increased expression of NLRP3, ASC, caspase-1 and IL-1β was found in colonic biopsies from CD and UC patients (Table 2) [161,162]. Interestingly, a recent study found upregulated NEK7, NLRP3, and caspase-1 expression, together with the pyroptosis-related inflammasome GSDMD gene, in intestinal tissue from UC patients [163]. The authors further showed that NEK7 knockdown in the intestinal epithelial cell line MODE-K reduced the LPS-induced pyroptosis, suggesting a NEK7-NLRP3-dependent pyroptotic mechanism in the pathogenesis of IBD. Next to NLRP3, increased mRNA levels of other inflammasome sensors, including NLRP1, NLRC4, NLRP6, NLRP12 and AIM2, were observed in colon biopsies from CD and UC patients [195]. In the case of NLRP6, a 131-fold and 3.9-fold increase in expression was reported in ileal CD and colonic CD patients, respectively.

### 3.2. Autoimmune Diseases

Autoimmune disorders are characterized by sustained autoreactive immune responses that result in organ damage and the production of autoantibodies. Although the exact pathologic mechanisms of most autoimmune disorders remain unclear, many studies have implicated inflammasomes in the pathogenesis.

#### 3.2.1. Rheumatoid Arthritis

Rheumatoid arthritis (RA) causes inflammatory degradation of cartilage and joint destruction and has a prevalence of 0.27% worldwide [196]. Of the known inflammasomes, NLRP1 and NLRP3 have been implicated in RA pathogenesis. Gene expression studies have revealed increased ASC, NLRP3, and caspase-1 expression in monocytes from RA patients (Table 2) [166], and higher intracellular levels of inflammasome components, including NLRP3, ASC, active caspase-1 and pro-IL-1β, as well as increased secretion of IL-1β was detected in patient monocytes [166,197]. In neutrophils, caspase-1 mediates IL-18 release independent of NLRP3 activity, suggesting that caspase-1 activity in neutrophils during RA is mediated by other inflammasomes or through the action of neutrophil proteolytic enzymes, such as metalloproteases, serine proteases and cathepsins, that may activate caspase-1 [198]. Other findings from the same study showed a positive correlation between RA severity and serum IL-18, but not IL-1β levels suggesting that IL-18, rather than IL-1β, contributes to disease progression. This may explain why, although the IL-1 receptor antagonist anakinra is widely used for the treatment of this disease, several clinical studies found poor effectiveness of direct IL-1β inhibition for the treatment of RA [199].

The role of NLRP1 in RA is less well understood. In Han Chinese, Sui et al., showed that upregulated NLRP1 due to *NLRP1* polymorphisms was associated with an elevated risk of RA [164]. However, the role of *NLRP1* polymorphisms in RA susceptibility is controversial, as other studies found no such correlation [165,200]. Nevertheless, synovial cells from RA patients show increased NLRP1, ASC and caspase-1 expression and high IL-1β secretion in vitro; here, NLRP1 activation and IL-1β secretion could be abrogated by inhibiting the P2X4 receptor [165]. P2X4 receptor inhibition could have promising therapeutic potential; however, no clinical trials with P2X4 receptor antagonists for the treatment of RA have been conducted. In contrast, several phase I and II trials did not find any significant efficacy of P2X7 receptor antagonists in the treatment of RA, suggesting that, at least, P2X7 is not a useful therapeutic target for RA [201,202].

#### 3.2.2. Sjogren’s Syndrome

Sjogren’s syndrome (SS) is a rheumatic disease with heterogeneous clinical manifestations ranging from mild dryness of the mouth and eyes to more severe systemic complications including interstitial lung disease, tubulointerstitial nephritis and non-Hodgkin’s lymphoma [203]. Studies of human SS samples found increased P2X7 receptor expression in the salivary glands and peripheral blood mononuclear cells of patients compared to controls [167,168], which correlated with increased NLRP3, ASC, caspase-1 and pyrin expression in the salivary glands and elevated IL-18 levels in the saliva of SS patients (Table 2). Elevated circulating levels and salivary gland expression of IL-18 also positively correlate with disease activity, and the production of autoantibodies and lymphoid infiltrates [204]. This correlation between P2X7 receptor expression, and the production of inflammasome components in SS, suggests an ATP-dependent mechanism of inflammasome activation. Indeed, P2X7 receptor activation by ATP can induce inflammatory responses in salivary gland epithelial cells in vitro [205], and salivary gland inflammation in a mouse model of autoimmune exocrinopathy [205]. Consequently, P2X7 receptor blockade can inhibit IL-1β release from salivary gland epithelial cells and reduces inflammation in this mouse model [205]. In addition, treating SS patient monocytes with ATP upregulates P2X7 receptor expression compared to that seen in control cells [206]. Importantly, such elevated expression of the P2X7 receptor, as well as NLRP3, ASC and caspase-1, predicts non-Hodgkin’s lymphoma development in SS patients [207]. Given this clear evidence on the involvement of the P2X7 receptor/NLRP3 axis in SS, P2X7 receptor poses an interesting therapeutic target for the treatment of the disease.

Besides ATP-dependent inflammasome activation, NLRP3 and AIM2 activation by cell-free DNA (cfDNA) has been detected in SS patients [169]. cfDNA accumulates in the peripheral blood of SS patients due to an impaired clearance of apoptotic cells and necrotic cell debris, and reduced DNase activity [169,170]. Notably, SS patients with established (or at high risk of) lymphoma exhibit high cfDNA levels, resulting in NLRP3 activation in circulating monocytes and NLRP3 activation and pyroptosis in macrophages infiltrating the salivary glands [169]. Finally, low DNase I expression in salivary epithelial cells and AIM2 co-localization with damaged genomic DNA in SS specimens, implies a role for defective cytosolic DNA degradation in the activation of AIM2 in SS [169].

#### 3.2.3. Systemic Lupus Erythematosus

Systemic lupus erythematosus (SLE) is characterized by multisystem inflammation with an overt IFN signature and autoantibody production against nuclear and cytoplasmic antigens [208]. Most of these antigens derive from apoptotic cells and neutrophil extracellular traps (NETs); as such, SLE severity strongly correlates with impaired degradation of NETs and anti-dsDNA antibodies [209]. Elevated IL-1β levels have been reported in SLE patient serum [210]. Mechanistically, it seems that anti-dsDNA antibodies can induce NLRP3-dependent IL-1β secretion by mononuclear cells and monocytes, thus explaining this elevation (Table 2) [171]. Others have found that NLRP3 activation is mediated via binding between anti-dsDNA antibodies and TLR4, inducing mitochondrial ROS production and K^+^ efflux [171,172]. Finally, Antichios et al. showed recently that AIM2 binds to NETs, thereby preventing its degradation, and the resulting nucleoprotein complex serves as an autoantigen that sustains IFN signaling [173].

### 3.3. Neuroinflammatory and Neurodegenerative Diseases

IL-1β and IL-18 have important roles in central nervous system (CNS) processes including cognition, learning, and memory [211]. IL-1β, mainly produced by microglia and astrocytes, has been linked to neural proliferation, differentiation, apoptosis, and long-term potentiation [212]. Elevated IL-1β and IL-18 levels seen after brain injury and in neurodegenerative diseases have also been implicated in their pathogenesis. In addition, cell death, including pyroptosis, promotes neuroinflammation and neural degeneration in multiple sclerosis, Parkinson’s disease and Alzheimer’s disease. In the following sections, we outline the involvement and pathological mechanisms of inflammasome activation in some of the most common neurological and neurological diseases.

#### 3.3.1. Alzheimer’s Disease

Accumulating data suggest an important role for inflammasome activation in the pathogenesis of Alzheimer’s disease (AD). When stimulated with LPS and synthetic Amyloid-β42 (Aβ), AD patient monocytes exhibit increased expression of the proteins involved in the assembly, activation, and downstream effectors of inflammasomes, including NLRP1, NLRP3, ASC, caspase-1, -5 and -8, IL-1β and IL-18 [174]. Furthermore, caspase activity in the brains and IL-1β concentrations in the cerebrospinal fluid of AD patients is generally increased compared to those in healthy controls [213]. High IL-1β expression is also detected in the microglia surrounding Aβ plaques in affected patients [214]. Indeed, activated microglia can release IL-1β upon in vitro stimulation with Aβ [177,215] in an NLRP3- and ASC-dependent manner [177]. Recent findings showed that astrocytes have functional NLRP3 inflammasomes and can also produce IL-1β upon Aβ stimulation, again, in an NLRP3- and ASC-dependent manner [178,179].

ASC contributes to AD pathogenesis through the formation of ASC specks, which are normally found within the microglia but can be released extracellularly upon microglial pyroptosis. Extracellular ASC rapidly binds Aβ peptides, thereby facilitating their aggregation [216], and such aggregates have been detected in brain samples from AD patients.

NLRP1 has also been implicated in AD but its exact role remains unclear. For example, NLRP1 levels have been found to be increased more than 25-fold in the neurons of AD patients, which might relate to a K^+^/Ca^2+^ imbalance due to the neurotoxic effects of Aβ on ion channels [175,176]. Furthermore, Aβ neural stimulation in mice resulted in elevated caspase-1 activity, IL-1β secretion and neural pyroptosis, which was dependent on NLRP1 activity [217]. Congruently, chronic treatment of PC12 cells with Aβ induced NLRP1/caspase-1/GSDMD-dependent pyroptosis followed by the release of IL-1β and IL-18 [218]. Finally, elevated NLRC4 and ASC expression was detected in brain samples isolated from sporadic AD patients [180].

#### 3.3.2. Parkinson’s Disease

Patients with Parkinson’s disease (PD) have increased levels of inflammasome-associated proteins, including IL-1β and caspase-1 [219] that can cause neuroinflammation and subsequent damage to dopaminergic neurons [220]. Central to PD pathophysiology are the gradual loss of dopaminergic neurons in the substantia nigra pars compacta and the accumulation of intraneural aggregates of fibrillar α-synuclein (α-syn), known as Lewy bodies. Similar to Aβ in AD, fibrillar α-syn induces IL-1β release in monocytes and microglia in a process dependent on NLRP3-mediated caspase-1 activity [181,182]. NLRP3 activation in these cells depends on the phagocytosis of fibrillar α-syn, resulting in increased ROS production and cathepsin B release into the cytosol. Moreover, this effect requires α-syn binding to TLR2, as evidenced by the blockade of α-syn-mediated NLRP3 activation and subsequent IL-1β release in human monocytes following anti-TLR2 antibody treatment [182]. Interestingly, Wang et al. found that caspase-1 colocalizes with Lewy bodies in PD patients and, more importantly, detected direct cleavage of α-syn by caspase-1, generating aggregation-prone fragments that were toxic to neuronal culture [221]. Together, these findings suggest a disease model with a dual effect of the α-syn/inflammasome interaction. In a first step, α-syn seems to activate the NLRP3 inflammasome, leading to caspase-1 activation and subsequent inflammatory cytokine release. The exact contribution of these inflammatory cytokines to PD pathophysiology needs further investigation. In a second step, activated caspase-1 is suggested to cleave α-syn, resulting in increased α-syn aggregation that potentially aggravates the disease. Given the central role of caspase-1 in this disease model, caspase-1 inhibition could be a potentially interesting therapeutic target for PD.

#### 3.3.3. Multiple Sclerosis

Multiple sclerosis (MS) is a common chronic inflammatory disease of the CNS characterized by defects in the blood–brain barrier [222] that permit local activation of microglia and astrocytes and the infiltration of immune cells to the brain from the periphery. The result is multifocal inflammation, demyelination, oligodendrocyte loss and neurodegeneration. Inflammasomes might mediate MS pathogenesis by contributing to inflammatory demyelination. Both infiltrating macrophages, microglia and astrocytes accumulate at sites of active demyelination and neurodegeneration [223]. These cells show increased NLRP3, ASC, caspase-1 and IL-1β expression, which significantly decreases in chronic, inactive MS lesions [183]. Recent research has revealed a pivotal role of *NLRP3* polymorphisms in MS [224], with gain-of-function variants linked to MS susceptibility and severity [225,226]. Notably, CAPS patients often develop MS later in life, supporting the role of NLRP3-mediated inflammation in MS [227,228].

In a recent study with a small cohort (*n* = 14) of MS patients, NLRP3, ASC, caspase-1, IL-1β and IL-18 were found to be significantly upregulated in postmortem CNS tissue compared to non-MS controls [184]. Transcript levels of NLRP1, NLRP2, AIM2 and Pyrin were also upregulated, although this was not statistically significant. Moreover, this study reported for the first-time molecular evidence of GSDMD-mediated pyroptosis in both myeloid (macrophages and microglia) cells and myelin-forming oligodendrocytes in the CNS of MS patients. These observations could be reproduced in vitro by exposure of human microglia and oligodendrocytes to inflammatory stimuli. Interestingly, caspase-1 inhibition by the small-molecule inhibitor VX-765 could strongly reduce the secretion of IL-1β and pyroptosis of microglia in vitro and reduced the expression of inflammasome- and pyroptosis-associated proteins in the CNS of a murine MS model. These observations offer a new perspective on the pathogenesis of MS and suggest a potential new treatment option for MS.

The roles of other inflammasomes in human MS are not well understood. NLRC4 was found abundantly expressed in lesions and astrocyte-rich regions from human brain tissue of three MS patients [179]. Moreover, NLRC4 has been detected in regions of demyelination, suggesting a role in MS pathology. A loss-of-function mutation in *NLRC4* was associated with reduced IL-18 production and a beneficial response to IFN-β treatment—the main treatment for MS [225]. IFN-β treatment also significantly decreases IL-1β, NLRP3 and AIM2 expression in MS patients, suggesting that IFN-β might improve MS symptoms by decreasing inflammasome-dependent IL-1β production. Finally, NLRP1 variants have also been implicated in MS. For example, Maver et al. identified a potential causative homozygous mis-sense variant in *NLRP1* in a familial form of MS [229]; however, these findings remain uncorroborated and additional research is needed to ascertain the role of *NLRP1* variants in familial MS [230].

## 4. Inflammasomes as Therapeutic Target for Inflammatory Diseases

Currently, immunosuppressive and anti-inflammatory treatment with cyclosporine, steroids, methotrexate, and general anti-TNF-a therapy is commonly used to treat severe cases of inflammatory diseases [231]. However, such treatments dampen the immune response but do not target disease-specific pathological mechanisms. Moreover, these treatments might interfere with the proper induction of protective immune responses. Therefore, it is essential to thoroughly understand individual inflammasome mechanisms, as this would allow aberrant inflammasomes to be targeted without dampening broader immune responses needed to fight infections and inflammatory insults.

The first-generation drugs targeting the inflammasome pathway target the downstream IL-1β cytokine and IL-1 receptor signaling. Canakinumab, a monoclonal anti-IL-1β antibody, is currently approved for treating several forms of arthritis, CAPS, and FMF, and its efficacy in reducing cardiovascular events in atherosclerosis patients has been shown in the large CANTOS trial [149]. In addition, Anakinra, a recombinant IL-1 receptor antagonist (IL-1RA), is used as a first-line treatment for CAPS and FMF and as a secondary treatment for rheumatoid arthritis [232]. A lentiviral IL-1RA gene therapy approach for treating CAPS is currently under development [233]. Although these therapies already target inflammasome-specific effectors, they do not target individual inflammasomes and might still interfere with the proper induction of protective immune responses.

Second-generation drugs are designed to specifically inhibit individual inflammasomes. To date, few inflammasome inhibitory drugs are under clinical development, all targeting the NLRP3 inflammasome. Inflazome biotech, recently acquired by Roche, has completed a Phase I clinical trial with Somalix (IZD334; Clinical Trials Identifier NCT04086602) and Inzomalid (IZD174; Clinical Trials Identifier NCT04015076) to assess safety and tolerability in healthy individuals and preliminary efficacy in adult patients with CAPS. Both drugs are analogs of the NLRP3 inhibitor MCC950 and are believed to act similarly by interacting with the NLRP3 NACHT domain, thereby inhibiting ATP hydrolysis and inflammasome activation and formation [234]. NodThera has been granted patents on inflammasome inhibitors based on carbamoyl derivates. Positive results in the Phase I clinical trial were obtained with compound NT-0796, an orally available drug, showing good blood–brain barrier penetration, highlighting its potential for treating neurodegenerative diseases, such as Parkinson’s disease [235]. NT-0249, on the other hand, is a peripherally restricted NLRP3 inhibitor that shows positive interim results from its Phase I trial, supporting the potential therapy of peripheral inflammatory diseases using a single daily dose. The mechanism of action of these compounds is not known. However, they have been shown to inhibit NLRP3 activation and concomitant IL-1β release [236]. Phase 2/3 is ongoing with the drug dapansutrile (OLT1177; Olatec Therapeutics) to test the safety and efficacy of this oral NLRP3 inhibitor for the treatment of acute gout flares, a form of inflammatory arthritis (Clinical Trials Identifier: NCT05658575). The first results of this study are expected by the end of 2023. Dapansutrile is a nitrile derivative that is believed to inhibit NLRP3 ATPase activity and NLRP3-ASC interaction through covalent interactions with the NLRP3 protein [237]. In addition, dapansutrile may also regulate IL1B, IL6, IL17A, IL18, MMP3 and TNF expression, thereby reducing the chemotaxis and activation of inflammatory cells [238].

Although these clinical trials are promising and show great potential for inhibiting inflammasome activation, much progress still needs to be made. Especially, the development of inflammasome inhibitors other than NLRP3 is lagging behind.

## 5. Conclusions and Future Directions

Inflammasomes have essential roles in the innate immune response. Still, aberrant activation or gain-of-function mutations in inflammasome proteins can also contribute to the development and progression of various autoimmune and autoinflammatory diseases. Our knowledge of inflammasome biology and inflammasome activation mechanisms has advanced significantly over the past decade, which, in turn, is driving our understanding of their involvement in inflammatory diseases. To date, the best-described inflammasome is NLRP3 and most research regarding the involvement of inflammasomes in human diseases has focused on this inflammasome. As such, the mechanisms and involvement of other inflammasomes in human disease need to be clarified. Future research should investigate the specific mechanisms of activation and regulation of less characterized/explored inflammasomes, and how they contribute to inflammatory disorders. From a therapeutic perspective, understanding the unique mechanisms of each inflammasome would open new opportunities to selectively target aberrant inflammasomes in patients with inflammatory diseases.

Pyroptosis is garnering attention as a critical mechanism driving inflammatory responses and pathology. The role of GSDMD in pyroptosis has been extensively studied, but further research on the biochemical mechanisms of GSDMD-mediated pyroptosis is needed. The role of GSDMD-mediated pyroptosis in autoimmune and autoinflammatory diseases also remains unclear. Further studies should investigate the role of GSDMD in inflammatory conditions and assess the potential of GSDMD inhibition as an anti-inflammatory therapy.

Finally, the role of different inflammasome-dependent caspases in the pathology of human diseases needs to be further elucidated. Caspase inhibition might be a promising therapy for many inflammasome-driven pathologies and multiple caspase inhibitors have been developed as a potential treatment for different pathologies; however, only few have progressed to clinical trials. The major challenges regarding these inhibitors is their poor efficacy and target specificity and the development of adverse side-effects. In general, caspases play a central role in many cellular processes, including apoptosis. For example, in the context of PD, apoptosis in neurons is thought to be triggered by mitochondrial dysfunction; therefore, inhibition of apoptosis may prevent the removal of dysfunctional neurons ultimately leading to necrosis, which might exacerbate the disease. A better understanding of the various caspase functions in specific disease settings, as well as the development of inhibitors targeting specific caspases, is needed.

Despite substantial progress, current understanding of inflammasome biology is insufficient to exploit fully for the development of anti-inflammatory therapies. An in-depth understanding of how the different inflammasomes are activated, as well as their assembly and upstream signaling events, at both transcriptional and post-transcriptional levels, are key to identifying novel therapeutic targets. To overcome the current challenges, further studies on understanding the role of the different inflammasomes in inflammatory diseases is fundamental to effectively develop their inhibitors as a treatment for the different pathologies.

## Figures and Tables

**Figure 1 cells-12-01766-f001:**
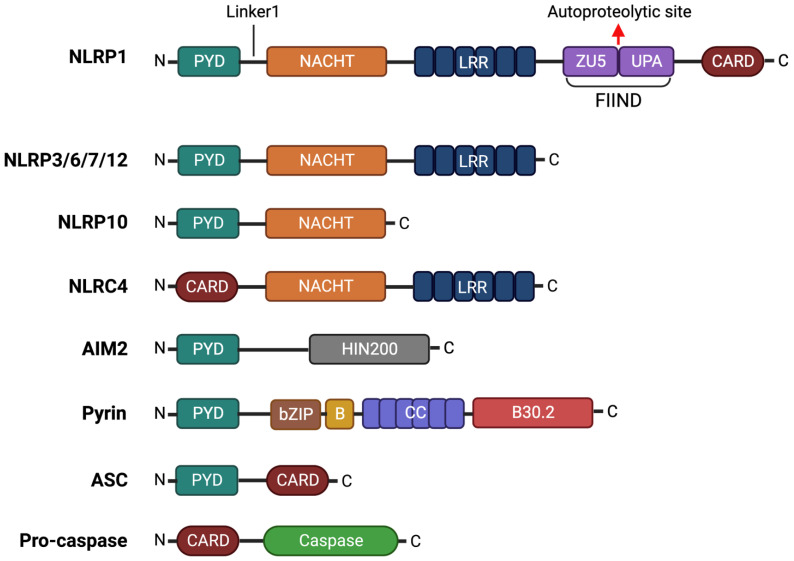
Domain organization of common inflammasome component proteins. NLRP1, NLRP3, NLRP6, NLRP7, NLRP10, NLRP12 and NLRC4 belong to the nucleotide-binding domain and leucine-rich repeat-containing receptor (NLR) protein family, which typically contains a central nucleotide-binding and oligomerization domain (NACHT) domain, an N-terminal Pyrin domain (PYD) domain and a C-terminal leucine-rich repeats (LRR) domain. NLRP1, NLRP10 and NLRC4 deviate from this typical NLR structure. In addition to the common structure, NLRP1 contains a C-terminal function-to-find domain (FIIND) and caspase recruitment domain (CARD), while NLRP10 lacks the LRR domain. The NLRC4 does not have the N-terminal PYD domain but, instead, has an N-terminal CARD domain. AIM2 belongs to the AIM2-like receptor (ALR) protein family and comprises an N-terminal PYD domain and a C-terminal HIN200 domain. Finally, Pyrin consists of an N-terminal PYD domain, a bZIP, B box, coiled-coil and an N-terminal B30.2 domain. Apoptosis-associated speck-like protein containing CARD (ASC) acts as an adaptor protein, consisting of a PYD and CARD, which connects the inflammasome sensor to pro-caspase through PYD and CARD interactions, respectively.

**Figure 2 cells-12-01766-f002:**
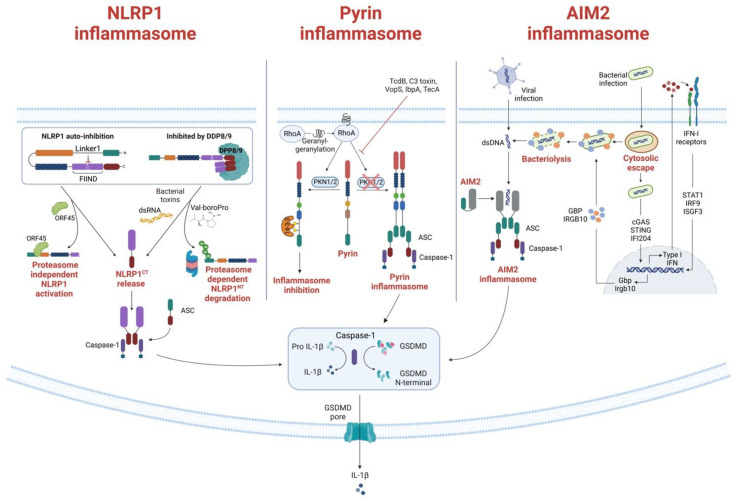
Formation and activation of Pyrin, NLRP1 and AIM2 inflammasomes. (**Left**) NLRP1 inflammasome formation. Under homeostatic conditions, NLRP1 is inactivated through auto-inhibition or by binding to the inhibitor dipeptidyl peptidases 8 and 9 (DPP8/9). The Kaposi sarcoma-associated herpes virus protein ORF45 is shown to bind to the Linker 1 region, lifting the auto-inhibition and DPP8/9 inhibition of NLRP1 and allowing NLRP1^CT^ to assemble the inflammasome. Another activation mechanism of the NLRP1 is through proteasomal degradation of the NLRP1^NT^. When bacteria or ubiquitin ligases ubiquitinate NLRP1, NLRP1 is directed to the proteasome, where NLRP1^NT^ is degraded, and NLRP1^CT^ is released for inflammasome assembly. The DPP8/9 inhibitor Val-boroPro can also direct proteasomal degradation of NLRP1 and subsequent release of NLRP1^CT^. (**Middle**) Pyrin inflammasome activation mechanism. RhoA activity is induced by geranylgeranylation (mevalonate kinase pathway). Pyrin is subsequently phosphorylated by the RhoA effector kinases PKN1 and PKN2, which then bind to the inhibitory protein 14-3-3. When PKN1/2 inhibiting substances are present—i.e., TcdB, C3 toxin, VopS—or when the mevalonate kinase (MVK) pathway is not functioning correctly, PKN1/2 is inactivated and reduced pyrin phosphorylation results in the release of mature IL-1 and IL-18 from the pyrin inflammasome. The creation of the gasdermin D (GSDMD) N-terminal fragment, which forms plasma membrane pores, further promotes the release of IL-1 and IL-18. (**Right**) AIM2 canonical and non-canonical activation. The canonical activation, which does not involve type I interferon (IFN) activation, is induced when dsDNA is directly recognized by AIM2, triggering the inflammasome formation. On the contrary, the non-canonical activation depends on IFN activity. It is principally involved in bacterial infections that escape the vacuoles, releasing a small amount of DNA that activates cyclic-GMP-AMP synthase and IFI204. Secreted IFN exits the cells and binds to IFN receptors, driving the downstream activation and inducing bacteriolysis which releases large quantities of bacterial DNA recognized by the AIM2 inflammasome. The activated AIM2 inflammasome drives the proteolytic maturation of IL-1β and IL-18 and the maturation of GSDMD, which induces pyroptosis.

**Figure 3 cells-12-01766-f003:**
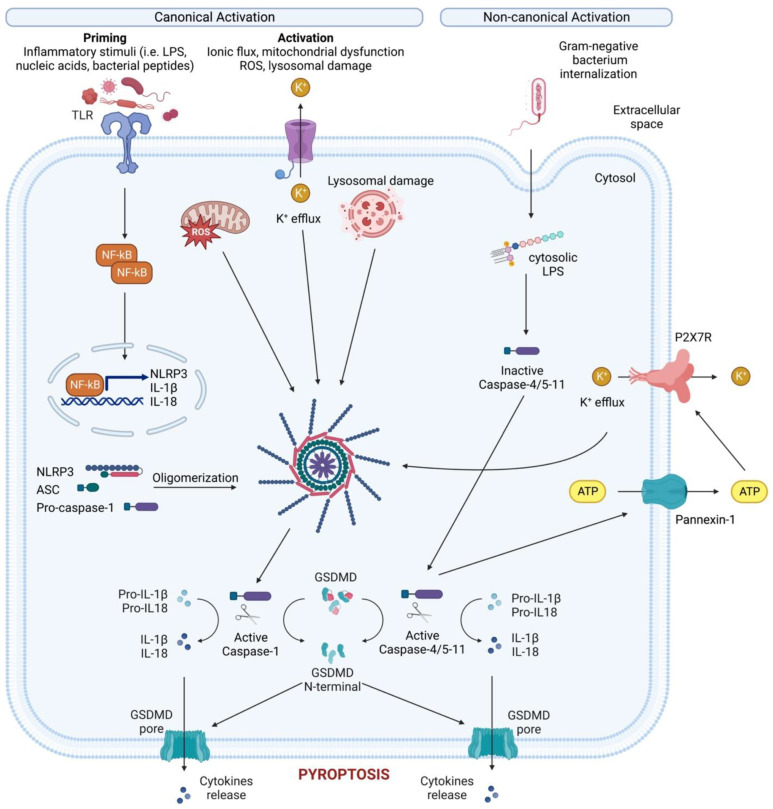
Canonical and non-canonical NLRP3 inflammasome activation. Canonical NLRP3 inflammasome activation requires two steps: the priming step and the activation step. In the priming step, TLR stimulation induces the transcription and expression of NLRP3 and pro-IL-1 through NF-κB. Subsequently, various PAMPs and DAMPs induce the activation step by initiating numerous molecular and cellular events, including K+ efflux, mitochondrial dysfunction, reactive oxygen species (ROS) release, and lysosomal disruption. The NLRP3-dependent self-cleavage and activation of pro-caspase-1 self-cleavage and activation leads to the maturation of the pro-inflammatory cytokine’s interleukin 1 (IL-1) and interleukin 18 (IL-18). Additionally, gasdermin D (GSDMD) is cleaved by activated caspase-1, releasing its N-terminal domain, which then integrates into the cell membrane to create pores. These pores allow the release of cellular contents, including IL-1 and IL-18, and trigger pyroptosis, a form of inflammatory cell death. The non-canonical NLRP3 inflammasome is activated by cytosolic LPS, which directly interacts with caspase-4/5 in human (caspase-11 in mice). This interaction results in the autoproteolysis and activation of these caspases. The activated caspases subsequently open the pannexin-1 channel, allowing ATP release from the cell and activating the P2X7R, causing K+ efflux, canonical NLRP3 activation and the maturation of IL-1 and IL-18. In addition, activated caspase-4/5-11 cleaves GSDMD to cause membrane pore formation and pyroptosis, contributing to the release of IL-1 and IL-18.

**Table 1 cells-12-01766-t001:** Common symptomatic NLRP1, NLRP3, NLRP12, NLRC4 and MEFV gene variants linked to the corresponding disease according to the Infevers database (https://infevers.umai-montpellier.fr/web/index.php accessed on 4 June 2023). NLRP1-associated autoinflammation with arthritis and dyskeratosis (NAIAD), familial cold autoinflammatory syndrome (FCAS), Muckle–Wells syndrome (MWS), neonatal-onset-multisystem inflammatory disease (NOMID), autoinflammation with infantile enterocolitis (AIFEC), pyrin-associated autoinflammation with neutrophilic dermatosis (PAAND).

Gene	Disease	Mutations Linked to Disease Phenotype
*NLRP1*	NAIAD	A59P, R726W, L813P, P1214R and L1214L
*NLRP3*	FCAS	C259W, L305P, L353P, T436A, A439V, E525K, Y563N, E627G, M659K
	MWS	R170S, R260L, L264V, D303A, E311K, H312P, R325W, T348M, A352V, K355T, A439T, F523C, E567K, E567A, G569R
	NOMID	R260P, V262A, L264F, L264H, L264R, D303H, E304K, G307S, G307V, F309S, G326E, A352T, E354D, H358R, A374D, T405P, M406V, M406I, T436P, T436N, A439P, F443L, N477K, F523Y, E525V, F566L, K568N, G569A, Y570C, Y570F, L571F
*NLRP12*	FCAS2	R284X, D294E, H304Y, W408X, S578G, L591M, L710P, R753H, N940S, S979G, R754H, F402L, G448A
*NLRC4*	FCAS4	H443P, T177A
	AIFEC	G172S, T177S, T337S, T337N, L339P, V341L, V341A, H443P, H443Q, W655C, Q657L, delexon5, Q880E
*MEFV*	FMF	K25R, R39G, E84K, A89T, Q97X, E167D, 606_621dup, K224del, S242R C > G, T267I, P313H, R354W, L372P, L384P, D389V, L396F, E403K, Y471X, F479L, R501C, S503C, 1611-1G > C, S650Y, G668R, M680L, M680V, M680IGA, G687D, Y688F, Y688X, I692DEL, M694V, M694L, M694DEL, M694K, M694I, K695N, V726A, F743Y, Q753H, R761H, N766H, P769A, Q778Sfs*4
	PAAND	S242G, S242R C > A, E244K, S363N

## Data Availability

Not applicable.

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
