# Peer review of "Inflammasomes: Mechanisms of Action and Involvement in Human Diseases"

_cells, 2023, doi:10.3390/cells12131766_

Round 1

Reviewer 1 Report

The authors did a very good job summarizing the role of the inflammasomes in the human diseases. However, after analyzing the text in a text similarity detection tool. I found that 33% of the text is similar to previous works. I ask the authors to rephrase the text in such a way that the similarity falls at least below 30%.

Additionally, the authors do not mention the expressing tissues/cells of the different inflammasomes. I ask the authors to please add in an explicit way the different tissues or cell types where each inflammasome is expressed. I also believe that, although, the figures clearly show how the inflammasomes are activated, the addition of molecules involved in it activation will make it better for the readers.  I please ask the authors to include a table where they summarize the different cell types where each inflammasome is expressed as well, as the identified activators, with the corresponding references.

The diseases part of the review is acceptable; however, I think the addition of a figure will also make it better and will enhance the quality of the review. Please consider it, or discuss why you believe it is not a good idea.

Finally, in the last section the authors mention very briefly some treatments that have been used, proposed/tested for the inflammasomes. I think the addition of a section where these treatments are described more in detail and also mechanistically, and also in the context of diseases will improve the review. I ask the authors to add this.

Reviewer 2 Report

In this study, Bulté et al. review the mechanistic role of inflammasomes in different human diseases. Inflammasomes are certainly a hot topic in the field of inflammatory diseases and pose attractive therapeutic targets, making this a timely review. The article is well-written and details the critical aspects of inflammasome activation and its roles. Furthermore, despite recent literature reviews on this topic, the article has updated literature and novel perspectives.

The authors need to address a few concerns before final acceptance:

  1. The authors discuss NLRP3 polymorphisms and the disease relevance. However, they do not mention specific polymorphisms and prevalence. I recommend adding a section/subsection discussing the genetics component of inflammasomes and disease relevance.
  2. Add a section (with figure if not redundant with the following sections) on inflammasome assembly.
  3. One table summarizing different NLRPs and another table summarizing the role of inflammasomes in different diseases would make the reading easier.
  4. Increase the font size of the text in Figures 2 and 3.
  5. Add an appendix of abbreviations used in the text.

Reviewer 3 Report

To Author:

Inflammasomes have essential roles in the innate immune response, but aberrant activation or gain-of-function mutations in inflammasome proteins can also contribute to the development and progression of various autoimmune and autoinflammatory diseases. In this review paper, the authors introduced the inflammasome structure and mechanisms of action and detailly summarized the roles of inflammasomes in different diseases, including inflammasomopathies, atherosclerosis, psoriasis, rheumatoid arthritis, sjogren’s syndrome, systemic lupus erythematosus, alzheimer’s disease, parkinson’s disease and multiple sclerosis. I considered this review paper to be comprehensive. However, I have several suggestions before it can be accepted.

 Comments:

(1) The inflammasome also plays an important role in the pathogenesis of inflammatory bowel disease and cancer. However, the authors did not introduce the research status of inflammasome in inflammatory bowel disease and tumor in this review.

(2) Multiple references are not properly cited, missing page numbers or journal name, such as 7, 12, 187 and 204.

Round 2

Reviewer 1 Report

The authors have correctly addressed the issues present in the submitted manuscript. Unfortunately, the assessment for plagiarism still shows a level above 30% which is deemed unacceptable. The authors mentioned the used of turnitin as a tool (turnitin is considered more of a tool for student works rather than scientific research); however, I recommend the usage of other tools instead, which are accepted as plagiarism detection tools for researchers.

Author Response

-

Reviewer 2 Report

The authors have addressed my concerns satisfactorily.

Author Response

We thank the reviewer for the positive comment.